# The potential of digital behavioural tests as a diagnostic aid for psychosis

**Piotr Słowiński**[1]*, **Alexander White**[2], **Sian Lison**[3], **Sarah Sullivan**[4], **Tobit Emmens**[3], **Philip Self**[3], **Jane Wileman**[5], **Anke Karl**[2], **Krasimira Tsaneva-Atanasova**[1,6]

**1** Translational Research Exchange @ Exeter, Living Systems Institute, Department of Mathematics and Statistics, Faculty of Environment, Science and Economy, University of Exeter, United Kingdom, **2** Department of Psychology, Faculty of Health and Life Sciences, University of Exeter, United Kingdom, **3** Research & Development Department, Devon Partnership NHS Trust, Exeter, United Kingdom, **4** Faculty of Health Sciences, Bristol Medical School, University of Bristol, United Kingdom, **5** Specialist Team for Early Psychosis, Devon Partnership NHS Trust, Exeter, United Kingdom, **6** EPSRC Hub for Quantitative Modelling in Healthcare University of Exeter, Exeter, United Kingdom

* p.m.slowinski@exeter.ac.uk

## Abstract

Timely interventions have a proven benefit for people experiencing psychotic illness. One bottleneck to accessing timely interventions is the referral process to the specialist team for early psychosis (STEP). Many general practitioners lack awareness or confidence in recognising psychotic symptoms or state. Additionally, referrals for people without apparent psychotic symptoms, although beneficial at a population level, lead to excessive workload for STEPs. There is a clear unmet need for accurate stratification of STEPs users and healthy cohorts. Here we propose a new approach to addressing this need via the application of digital behavioural tests. To demonstrate that digital behavioural tests can be used to discriminate between the STEPs users (SU; n = 32) and controls (n = 32, age and sex matched), we compared performance of five different classifiers applied to objective, quantitative and interpretable features derived from the 'mirror game' (MG) and trail making task (TMT). The MG is a movement coordination task shown to be a potential socio-motor biomarker of schizophrenia, while TMT is a neuropsychiatric test of cognitive function. All classifiers had AUC in the range of 0.84–0.92. The best of the five classifiers (linear discriminant classifier) achieved an outstanding performance, AUC = 0.92 (95%CI 0.75–1), Sensitivity = 0.75 (95% CI 0.5–1), Specificity = 1 (95%CI 0.75–1), evaluated on 25% hold-out and 1000 folds. Performance of all analysed classifiers is underpinned by the large effect sizes of the differences between the cohorts in terms of the features used for classification what ensures generalisability of the results. We also found that MG and TMT are unsuitable in isolation to successfully differentiate between SU with and without at-risk-mental-state or first episode psychosis with sufficient level of performance. Our findings show that standardised batteries of digital behavioural tests could benefit both clinical and research practice. Including digital behavioural tests into healthcare practice could allow precise phenotyping and stratification of the highly heterogenous population of people referred to STEPs resulting in quicker and more personalised diagnosis. Moreover, the high specificity of digital behavioural tests could facilitate the identification of more homogeneous clinical high-risk populations, benefiting research on prognostic instruments for psychosis. In summary, our study demonstrates

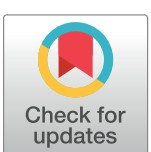

**Data Availability Statement:** The fully anonymised research data supporting this publication are openly available from: https://doi.org/10.17605/OSF.IO/RNZYS.

**Funding:** The research was supported by, EPSRC Impact Acceleration Account, Impact & Knowledge Exchange Award, Jean Golding Institute seed corn, Avon & Wiltshire Mental Health Partnership NHS Trust Research Capability Funding. PS was generously supported by the Wellcome Trust Institutional Strategic Support Award 204909/Z/16/Z. KTA gratefully acknowledges the financial support of the EPSRC via grant EP/T017856/1. The funders had no role in study design, data collection and analysis, decision to publish, or preparation of the manuscript.

**Competing interests:** The authors have declared that no competing interests exist.

that cheap off-the-shelf equipment (laptop computer and a leap motion sensor) can be used to record clinically relevant behavioural data that could be utilised in digital mental health applications.

## Author summary

Neuropsychiatric assessment and accurate diagnosis are notoriously challenging. Psychosis represents a classical example of this challenge where many at-risk of psychotic illness individuals (often very young) are misdiagnosed and/or inappropriately treated clinically. Our study demonstrates that combining digital tests with data analytics has potential for simplifying neuropsychiatric assessment. It shows that using measurements from trail making task and mirror game allows to differentiate between people accepted for assessment in specialist team for early psychosis (STEP) and controls with outstanding performance (AUROC > 0.9), while achieving 100% specificity (no false positive detections). The study shows feasibility of using cheap, portable equipment, assembled from off-the-shelf components, for collection of clinically relevant data that could be used to inform clinical decision making. Moreover, our study, with its state-of-the-art performance and interpretable results, demonstrate high potential of implementing digital batteries of behavioural tests in clinical practice. Such developments would not only help to stratify STEPs users but would facilitate rapid assessment for all people seeking care in early intervention services. This in turn would contribute to improving the quality of life and wellbeing of all help seeking individuals.

## Introduction

Psychosis is a severe mental illness characterised by loss of contact with reality and symptoms such as hallucinations, delusions and thought disorders. It can be one of the first symptoms of a range of serious and long-term mental disorders such as schizophrenia, affective and other psychoses. Developing psychosis in young adulthood is devastating and often disrupts the trajectory into healthy and independent adulthood; the mean age of onset is 22 years for men and 26 years for women [1]. People with serious mental illness die 15 to 20 years earlier than the general population [2].

Serious mental disorders are extremely expensive to treat, with presence of psychotic or affective symptoms being one of patient characteristics driving increase in hospital costs [3]. Their direct healthcare toll to NHS England have been estimated at £2.82 billion annually in 2019 [3]. While the most recent estimate puts the overall annual economic impact of schizophrenia and psychosis in England at £11.8 billion [4]. The higher overall economic impact includes reduced labour supply, premature mortality, reduced health-related quality of life, lost output, lost tax revenue, transfer payments, and unpaid care by family or friends.

Most risk factors for a poor outcome, such as gender or low socio-economic status, are difficult or impossible to alter. But people with psychosis have better outcomes if they are treated as soon as possible after their first symptom [5]. Early interventions can reduce the rate of relapse, risk of suicide, and number of hospital admissions [1,6]. They also significantly improve quality of life by enabling people to finish education and develop supportive networks outside the family of origin [1].

Early interventions are typically delivered by a specialist team for early psychosis (STEP) [7–12]. However, the referral process to STEPs is far from being optimal. Although most STEP

referrals are from primary care, many general practitioners lack awareness of high-risk symptoms or are not confident with recognising the psychotic state, both of which could lead to people not receiving the care they need [13,14]. On the other hand, although increasing the number of referrals has been shown to be beneficial at the population level, it also leads to increases in STEPs workload thus contributing to the pressure on the care system. The extra work is caused by higher number of assessments requested as well as a need for an increased engagement (including dedicated liaison practitioners) with primary care providers to identify and refer people experiencing, or at risk of, psychotic illness [9,15].

Here, we investigate if digital behavioural tests can be used as an effective tool that allows differentiation between people referred to STEPs and the general population, and if they show potential to facilitate and standardise the referral process. Specifically, we use data from a digital version of the trail making task (TMT), a standard method for assessment of cognitive function [16,17], and the mirror game (MG), a novel way of assessing socio-motor functioning (motor coordination and interpersonal synchronisation) [18,19]. Our choice of the tasks is based on a significant body of research showing that assessment of movement, behaviour and cognitive function allows to accurately differentiate between people with schizophrenia and general population [18,20–31]. In particular, motor and executive functions [18,30] as well as eye movements [24,27] were shown to hold promising diagnostic potential. In addition, deficits in motor coordination were recently shown to be markers of long-term clinical outcomes [31], while performance in TMT was shown to differ in people at clinical high-risk for psychosis who transitioned from those who did not transition to psychosis [32]. Moreover, this existing body of research ensures generalisability of the presented results from sample to population. In addition, our study demonstrates feasibility of using cheap off-the-shelf components (laptop computer with a plug-in sensor) for simplifying neuropsychiatric assessment and introducing standardised digital tests to clinical practice [20].

## Methods

### Study design and participants

The study was designed as a prospective, cross-sectional feasibility study in a group of service users accepted for an assessment for psychosis including people with first episode psychosis or assessed as being at risk of developing psychosis. Control cohort was recruited independently at the University of Exeter (UoE). Demographic and clinical characteristics of participants can be found in Table 1.

**Table 1. Demographic and clinical characteristics of participants.**

| | Service users (N = 32) | | Controls cohort (N = 32) | | Statistics |
|---|---|---|---|---|---|
| | **Mean** | **Min-Max** | **Mean** | **Min-Max** | |
| Age (years) | 28.8 | 18–58 | 24.2 | 18–54 | U = 394.5, p = 0.11 |
| Sex (male/female) | 16/16 | | 12/20 | | $Chi^2$ = 1.01, p = 0.31 (Pearson) |
| CAARMS (score, number of participants) | | 0, n = 16; 2, n = 4; 4, n = 12 | | | |
| CAPE-42 | | | 1.41 | 0.85–2.36 | |
| Anti-psychotic medication | 15 participants, < 4 months | | | | |

Note: U–Mann-Whitney statistic, $Chi^2$ –Chi-squared statistic, CAARMS–score of the comprehensive assessment of at risk mental state, CAPE-42 –Community Assessment of Psychic Experiences-42.

Service users (SU) were identified and recruited by Devon Partnership NHS Trust (DPT) and Avon and Wiltshire Mental Health Partnership NHS Trust (AWP). In total we recruited 32 participants, all of which were included in the analysis (we do not have data about number of screened participants, none of the participants dropped-out). The inclusion criteria were being accepted for an assessment for psychosis or risk of developing psychosis by a consultant psychiatrist or a trained specialist with experience in at-risk mental states. The exclusion criteria were:

1. Lacking capacity to provide informed consent for inclusion. The clinical team had an opportunity to assess the mental capacity at the CAARMS (comprehensive assessment of at-risk mental state (ARMS) [33] appointment before potential participants were approached to request consent to contact.

2. Insufficient understanding of English to follow the test instructions.

3. Any suspected organic cause of psychosis (i.e., head injury, epilepsy or dementia).

4. Taking antipsychotic medication for longer than 4 months before the start of the study.

Each participant was offered £10 for participating and if necessary, reimbursement for reasonable travel expenses (after producing a receipt). SU were recruited between 19/07/2018–23/05/2019. The study was reviewed by Research Ethics Committee (REC) and received approval from Health Research Authority (HRA) and Health and Care Research Wales (HCRW); IRAS (Integrated Research Application System) ID: 236262, REC reference: 18/SW/0065, protocol number 1718/26.

Control cohort (CC) was identified at UoE. In total we recruited 86 eligible participants, of which 43 played the same version of MG as SU and were used to identify the n = 32 CC matching by age and gender the SU as close as possible. Participants were volunteers recruited by personally approaching potential participants, putting posters around UoE campus and at Exeter's community centres, social media adverts, and snowball sampling. The exclusion criteria were:

1. Moderate, or more severe, symptoms of depression, assessed by means of Patient Health Questionnaire-9 (PHQ-9) [34]. For ethical reasons we excluded the question concerning thoughts of suicide and self-harm. All participants scoring above 9, indicating at least moderate levels of depression, were signposted to several sources of support.

2. A diagnosis of depression, an anxiety disorder or schizophrenia.

3. Taking any psychopharmacological medication. Participants who indicated that they were having difficulties with mental health were directed to the UoE wellbeing centre.

4. Suffering from seizures.

5. English not being one of their first languages. This criterion was introduced to try to minimise the chances of misinterpretations due to the extensive use of questionnaires in the study.

CC was additionally screened using Community Assessment of Psychic Experiences-42 (CAPE-42) [35]; question 14, which asks about suicidal ideation and loads onto the depressive subscale, was excluded for ethical reasons. Each participant was offered £5 or one course credit for participating. CC was recruited by AW between 25/05/2018–26/11/2018 as part of his Master's degree project. The recruitment of the CC was approved by University of Exeter, College of Life and Environmental Sciences (CLES), Psychology Ethics Committee, eCLESPsy000568 v2.1.

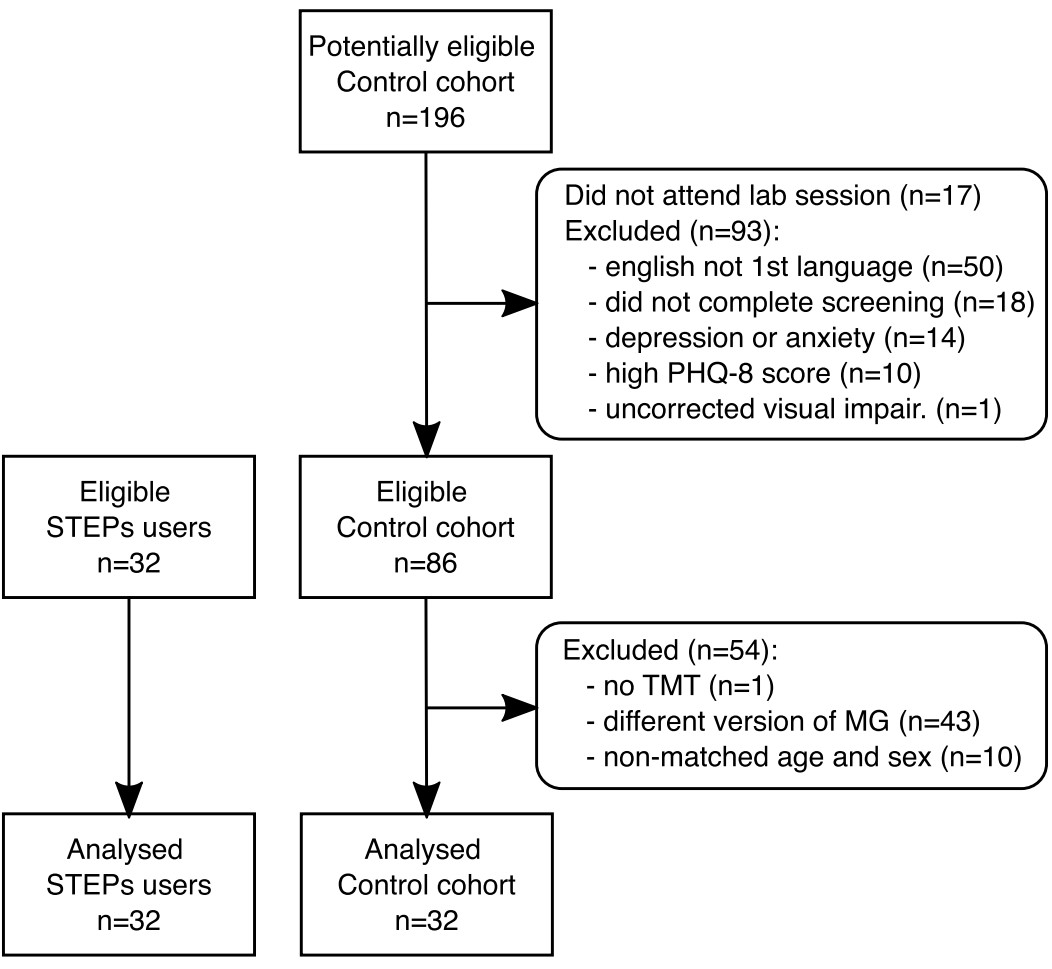

**Fig 1. Participant flow chart.**

Participant flow chart is presented in Fig 1. All participants gave written informed consent prior to the study.

Further information about the study can be found at https://www.hra.nhs.uk/planning-and-improving-research/application-summaries/research-summaries/movement-and-perspective-taking-as-a-diagnostic-aid-for-psychosis/. Full study protocol can be accessed at http://hdl.handle.net/10871/132205.

## Mirror game

The mirror game (MG) used in this study was based on the algorithm described by Zhai and colleagues [36] and followed closely our earlier work on establishing socio-motor markers of schizophrenia [18]. MG is a movement task that can be used to assess socio-motor functioning (motor coordination and interpersonal synchronisation) [18,19]. We used two MG tasks. The first task was a Solo game, where participants were asked to move their hand freely in a horizontal direction. Participants were given the following instruction: "Please move your hand left and right, create an interesting motion and enjoy playing." The second task was a Leader-Follower game. In the second task an animated image of a robot appeared on the screen. The animation showed the robot controlling its own dot. The dot moved horizontally according to

a pre-generated movement pattern. Participants were tasked with following the dot's movement as closely as possible whilst it was on screen. Participants were given the following instruction: "Please try to follow the movement of an animated robot as accurately as you can." During the Leader-Follower game the robot was also presenting parametric positive social feedback (smiling) as described by Cohen and colleagues [37].

The two tasks were grouped into one session. The session consisted of the Solo game, three repetitions of the Leader-Follower game and another Solo game. Each game lasted for one minute. The session was repeated three times. Participants were free to take breaks between the games and sessions. Each Leader-Follower game used a different pre-generated movement pattern. Patterns were the same for each participant. We excluded the 1st Solo game and the 1st Leader-Follower game to allow participants to get familiar with the task. The SU participants were sitting in front of a 17" diagonal laptop computer (1100x680 pixels screen resolution), the CC was using a 23" diagonal computer monitor; the image displayed on the computer monitor was scaled down to use the central 17" diagonal part of the monitor and have the same resolution as the laptop display. Movement of the hand was recorded using a leap motion sensor [38] and displayed as a dot on the screen. Participants used their dominant hand to control the horizontal position of a dot on the screen. The computer set-ups were different in the two groups due to the need of collecting the data simultaneously in multiple locations and additional research goals for the experiments with the CC that are not a part of the presented analysis.

In our analysis we used the recorded position of the participant hand (Solo and Leader-Follower) and the trajectory of the movement generated by the computer (Leader-Follower). Recorded position data is in arbitrary units in the range [-0.5, 0.5]; variable sampling rate, 90–140 Hz (S) and 40-70Hz (Leader-Follower). Pre-processing included:

- resampling to 100Hz with linear interpolation,
- low pass filtering with 5 Hz cut-off done using phase preserving Butterworth filter of degree 2,
- omitting the first and last 5s of the recording,
- estimation of movement velocity, using a fourth-order finite difference scheme.

### Trail-making task

The trail-making task (TMT) [16,39] is a valid, public domain test of visual attention, working memory and executive control [40]. It has two parts, which were alternated. In each part, participants must click on 25 dots in a specified order as quickly and accurately as possible. The visual attention part (TMT A) had participants click numbers in ascending order, 1–25. The executive control part (TMT B) had participants alternate between clicking on numbers and letters, both in ascending order (1-A-2-B-3-C etc.). Participants completed each part three times, alternating between TMT A and B, starting with TMT A. Only the last two repetitions were included in the analysis. We excluded the 1st repetition to allow participants to get familiar with the task. We used a digital version of the task implemented in PEBL: The Psychology Experiment Building Language [41]. In the original study protocol the TMT was used as a non-diagnostic attention measuring task. It was retrospectively included in the analysis after literature review [37,42–44] and data analysis indicated that including participants' performance in this task could be beneficial for differentiating between the SU and CC.

For analysis we used the times between each individual mouse click made by the participant, we also used times between mouse clicks made on the correct targets.

## Testing procedure

The stages of the research session are presented in Table 2. Both tasks and examples of collected data are shown in Fig 2.

## Sample size

For the feasibility study we recruited as many eligible SU as possible for the duration of the study. We approached all eligible participants that had been identified as appropriate for assessment for risk of psychosis by the Specialist Teams for Early Psychosis. Convenience sampling allowed us to proceed with the study as quickly as possible and assess what are feasible sample sizes for future research. Sample size of the CC was driven by the research objectives of AW's Master's degree project.

## Features extracted from data

The selection of features for classification was informed by our earlier work [18], and modified to better fit the machine learning methodology employed in the current study. Instead of using distributions (histograms) as in the previous work, here we use a set of their descriptive statistics (e.g., mean, standard deviation or median) or point measures (e.g., power at 5Hz frequency). As previously, the data was concatenated or averaged across the repetitions. Before averaging or concatenating the movement data, we remove parts where participants' moves reached the edges of the sensor range (-0.5 or 0.5 value). The complete list of the features, and their description, is presented in Table 3.

# Results

## Classes

We used the group (CC or SU), as the primary classification outcome (predicted variable), binary classification of the full dataset. Additionally, we used CAARMS score (CAARMS > 0 – at-risk mental state, psychotic or CAARMS = 0 –neither) as classification outcome of an independent binary classification within the SU group. CAARMS is one of the gold standards in

**Table 2. Stages of the research session.**

| Stage | Description |
|---|---|
| 1. | Time for questions and signing informed consent. |
| 1a. | CC additionally answered a questionnaire (state/ moment PANAS) [45] and the finger-tapping test [46], not a part of the presented analysis. |
| 2. | 1st MG session (excluded from analysis): |
| | Solo x 1–1 minute of participant's own movement; |
| | Leader-Follower x 3 –three repetitions of the participant following the avatar (1 minute each); |
| | Solo x 1. |
| 3. | Break (at least 1 minute). |
| 4. | 2nd MG session: Solo x 1; Leader-Follower x 3; Solo x 1. |
| 5. | Break (at least 1 minute). |
| 6. | 3rd MG session: Solo x 1; Leader-Follower x 3; Solo x 1. |
| 7. | Break (at least 1 minute). |
| 7a. | CC additionally answered a questionnaire (state/ moment PANAS), not a part of the presented analysis. |
| 8. | TMT: 3 x TMT A and B. (1st repetition excluded from analysis.) |
| 8a. | SU and CC were asked to answer a MG acceptability questionnaire, not a part of the presented analysis. |
| 8b. | CC continued with remaining part of the research session. |

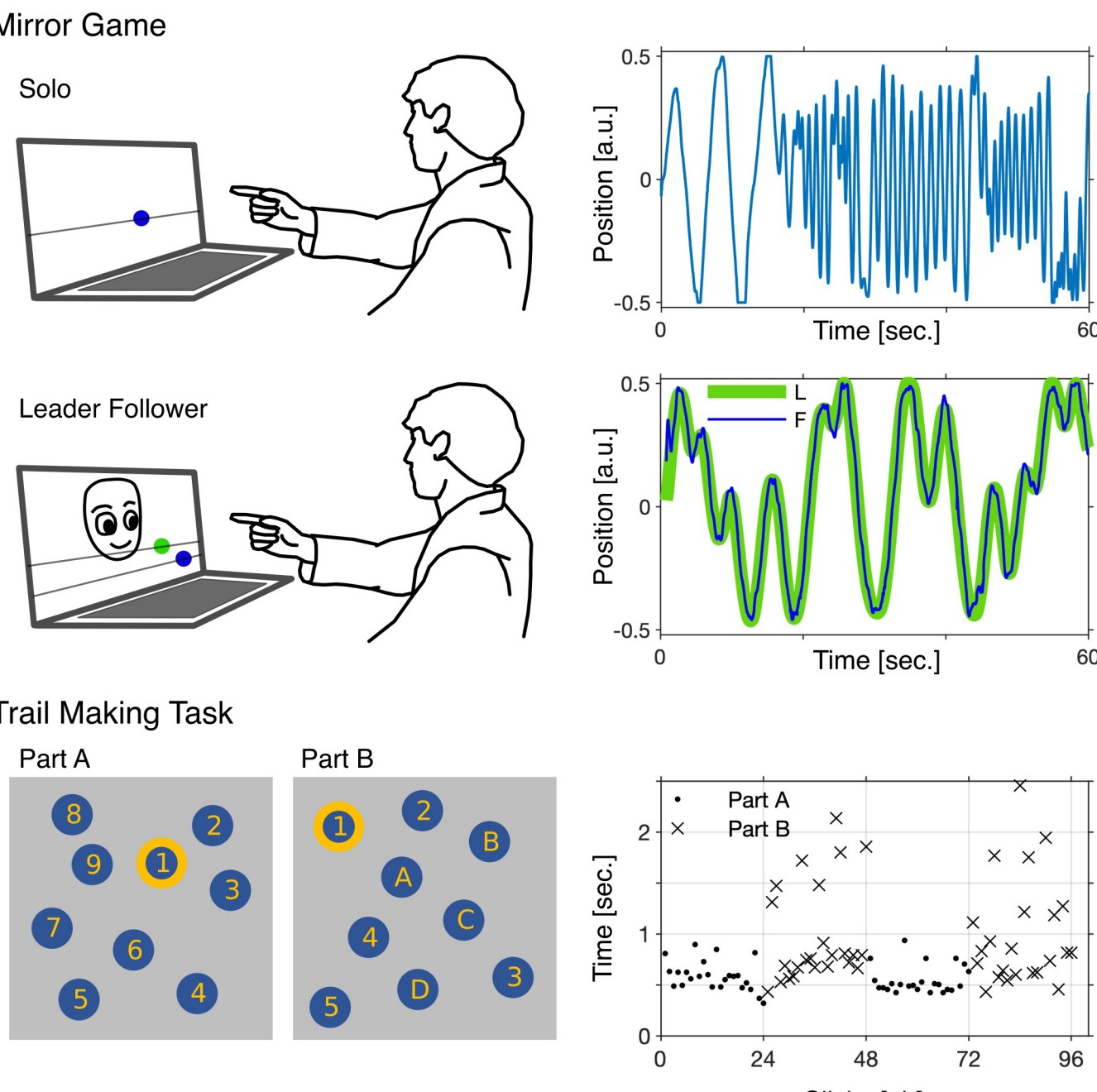

**Fig 2. Illustration of the Mirror Game (MG) and the Trail Making Task (TMT) together with examples of collected data.** In the MG participant sat in front of a computer with a connected leap motion sensor. In the Solo game (first row) the participant was instructed: "Please move your hand left and right, create an interesting motion and enjoy playing." We recorded the horizontal hand movement (blue). In the Leader-Follower game (second row) the participant was instructed: "Please try to follow the movement of an animated robot as accurately as you can." We recorded movement generated by the computer (leader (L), green) and movement of the participant (follower (F), blue). In the TMT participant sat in front of the same laptop computer but was using a computer mouse to complete the task. In the TMT (third row) the participant was asked to connect a set of 25 dots as quickly and accurately as possible (in order given by numbers (Part A) and alternating numbers and letters (Part B)). We recorded the time between each click a person made on the screen and analysed both parts together (Part A, dots; Part B, crosses). For the sake of clarity, we show simplified illustration with 9 dots instead of 25. Part of the MG illustration is based on the same source file as Fig 1 in [18], Fig 1 in [18] is distributed under the terms of the CC BY 4.0 license (CC BY 4.0).

**Table 3. List of features and data used to estimate them.**

| Features | Description of the data |
|---|---|
| TMT ICT mean<br>TMT ICT std<br>TMT ICT median | Inter-click times (ICT)–times in msec. between each mouse click made by the participant while completing parts A and B of TMT. As features we used mean, standard deviation (std) and median. ICTs are a natural extension of the completion times used typically as TMT measure. |
| TMT IoTCT mean<br>TMT IoTCT std<br>TMT IoTCT median | Inter-on-target-click times, (IoTCT)–times in msec. between on target (correct) mouse clicks made by the participant while completing parts A and B of TMT. As features we used mean, standard deviation and median. IoTCTs are a natural extension of the completion times used typically as TMT measure. |
| S GWS p pf<br>S GWS p mean<br>S GWS v pf<br>S GWS v mean | Global wavelet spectrum (GWS) based on position (p) or velocity (v) time-series from the solo (S) task. GWS is a normalised time-average of a wavelet power spectrum based on continuous wavelet transform [57]. We only consider frequencies in the 0.25 and 5Hz band. As features we used the frequency of the highest peak (pf) of the GWS, and mean GWS frequency estimated as: $\mu_{GWS} = \int_{0.25}^{5} \omega GWS(\omega) d\omega$.<br>We used GWS, rather than Fourier analysis, because it is better suited to characterise non-stationary time-series from the solo task [19,47]. |
| LF GWS pos. mean<br>LF GWS pos. 5HZ<br>LF GWS vel. mean<br>LF GWS vel. 5HZ | GWS based on position or velocity time-series of the human participant from the Leader-Follower (LF) task. As features we used the mean GWS frequency and GWS value (power) at 5Hz. We used GWS, rather than Fourier analysis, because it is better suited to characterise non-stationary time-series from the Leader-Follower task [19,47]. |
| LF RP frequency | Distribution of relative phase across wavelet frequency bands (limited to 1/15–2 Hz band), estimated as a circular mean over time of the phase of the wavelet cross-spectrum (WCS) [18,19,47] computed between the leader and follower positions time-series in the Leader-Follower task. As a feature we used the frequency at which the phase value drops below -pi/4. WCS allows to quantify relative phase (lag) between the leader and follower movements. |
| LF RP time mean<br>LF RP time std<br>LF RP time median | Relative phase estimated as a circular mean over frequencies (limited to 1/15–2 Hz band) of the phase of the WCS computed between the leader and follower positions time-series in the Leader-Follower task. As features we used mean, standard deviation and median. |

assessing service users at risk of or with first episode of psychosis. CAARMS was completed by a consultant psychiatrist or a trained specialist with experience in at-risk mental states of STEPs before completion of the MG and TMT tasks.

## Classification methods

We compared following classifiers: k-nearest neighbours (kNN), naïve Bayes (NB), support vector machines (SVM), bagged trees (BT), linear discriminant (LD) [48]. To find the k-nearest neighbours for kNN classifier we used a cosine distance to measure distances between the points in the n-dimensional feature space (each coordinate of the feature space corresponds to a single z-scored feature). Cosine distance is defined as $1-cos(\theta)$, where $\theta$ is the angle between vectors defined by coordinates given by the set of features and the origin of the coordinate system. We used a NB classifier as implemented in Matlab 2022b function fitcnb [49], a SVM classifier as implemented in Matlab 2022b function fitcsvm [50], a BT classifier as implemented in Matlab 2022b function TreeBagger [51] and a LD classifier as implemented in Matlab 2022b function fitcdiscr [52]. All classifiers were trained using default Matlab 2022b settings.

To avoid overfitting, we used only three out of the 18 features (Table 3), namely one from the 6 features estimated from the TMT data, one from the 4 features estimated from the MG

Solo task and one from the 8 features estimated from MG Leader-Follower task. To select the features, we used the value of Cliff's delta [53], a non-parametric measure of effect size. The set of classification features was based on the results from the training phase, meaning that it was selected separately for each training-testing split (fold).

Additionally, we analysed how performance of the kNN and LD classifiers depends on the number of features. We compared kNN with 3 (selected as described above) and all 18 features and LD with 2 (selected as described above but without MG Solo task), 3 (selected as described above) and all 18 features.

## Training and testing

To evaluate the performance of the classifiers, we used two training-testing splits. A 25% hold-out (HO) training-testing split and a leave-one-out (L1O) training-testing split (corresponding to 2% hold-out in our case). Parameters of the classifier and the three features used for classification were identified using only the training set. Hold-out data is used only for testing and is unseen by the classifier during the training.

In the 25% HO split we selected at random 25% of the data (8 out of 32 participants in each cohort). We train the classifier using the remaining 75% of the data (24 CC and 24 SU datasets). We used the 16 participants (8 CC and 8 SU datasets) unseen by the classifier during training to construct confusion matrix, and compute performance metrics. To estimate 95% confidence intervals of the classifier performance we repeated the 25% HO split 1000 times (1000 folds).

The leave-one-out (L1O) training-testing split used n-1 participants to train the classifier and 1 participant to test the model. L1O training-testing split allowed to test the methodology n times. The L1O split simulated situation where a new participant would be diagnosed using classifier based on all the data available prior to the arrival of the new participant. To construct confusion matrix and compute performance metrics we compared the original classes with the set of individual predictions of each of the L1O splits, i.e., we compared original class of the participant unseen by the classifier with the class predicted by the model trained using the other n-1 participants.

## Classification results

All tested methods allowed classification of the CC and SU participants with an excellent (0.8–0.9) [54] or outstanding (0.9–1) [54] accuracy, sensitivity, specificity and precision. The only exception being acceptable (0.7–0.8) [54] or poor (0.6–0.7) [54] sensitivity in few cases (see Table 4).

Comparing how classification results depend on the number of features we found that the performance of the kNN classifier was comparable when using 3 or 18 features. The kNN classifier using three features had higher AUC but lower specificity and precision than the kNN using all 18 features. The performance of the LD classifier showed stronger dependence on the number of features. It performed best using three features and its performance decreased when using two and 18 features.

Furthermore, all the analysed methods failed to differentiate between SU with and without at-risk-mental-state (CAARMS score of 0 and CAARMS score > 0); binary classification within the SU cohort using CAARMS score as group label, CAARMS = 0 vs CAARMS > 0. Since there were only 16 participants with CAARMS = 0 and 16 participants with CAARMS > 0 we only used the L1O training-testing split. See Table 4 for details.

To better understand the difference in performance of the proposed methodology in the two cases (CC vs. SU and SU CAARMS = 0 vs SU CAARMS > 0), we compared distributions of the features in the 3 groups. We did not find any statistically significant differences between

**Table 4. Results of using 8 classification methods: k-nearest neighbours (kNN), k-nearest neighbours with 3 features (kNN3), naïve Bayes, support vector machines (SVM), bagged trees (BT), linear discriminant (LD), linear discriminant with 3 features (LD3) linear discriminant with 2 features (LD2) and two types of training-testing split (HO and L1O) for classifying SU and CC and results of L1O training-testing split classifying SU with and without at-risk-mental-state (CAARMS = 0 vs. CAARMS > 0). n in Testing dataset column shows number of datasets from the indicated cohort used for testing (see also the Training and testing section).**

| | | Testing dataset | TN | FP | TP | FN | AUC | Accuracy | Sensitivity | Specificity | Precision |
|---|---|---|---|---|---|---|---|---|---|---|---|
| kNN | HO | CC, n = 8 | 8 | 0 | 6 | 2 | 0.89 | 0.88 | 0.75 | 1 | 1 |
| | | SU, n = 8 | (5–8) | (0–3) | (4–8) | (0–4) | (0.73–1) | (0.66–1) | (0.50–1) | (0.62–1) | (0.67–1) |
| | L1O | CC, n = 32 SU, n = 32 | 27 | 5 | 26 | 6 | 0.86 | 0.83 | 0.81 | 0.84 | 0.84 |
| | L1O | CAARMS = 0, n = 16 CAARMS>0, n = 16 | 3 | 13 | 5 | 11 | 0.21 | 0.25 | 0.31 | 0.19 | 0.28 |
| kNN3 | HO | CC, n = 8 | 7 | 1 | 6 | 2 | 0.91 | 0.88 | 0.75 | 0.88 | 0.88 |
| | | SU, n = 8 | (5–8) | (0–3) | (5–8) | (0–3) | (0.71–1) | (0.69–1) | (0.62–1) | (0.62–1) | (0.67–1) |
| | L1O | CC, n = 32 SU, n = 32 | 27 | 5 | 25 | 7 | 0.89 | 0.81 | 0.78 | 0.84 | 0.83 |
| | L1O | CAARMS = 0, n = 16 CAARMS>0, n = 16 | 5 | 11 | 10 | 6 | 0.39 | 0.47 | 0.62 | 0.31 | 0.48 |
| Naïve Bayes | HO | CC, n = 8 | 7 | 1 | 7 | 1 | 0.89 | 0.88 | 0.88 | 0.88 | 0.88 |
| | | SU, n = 8 | (5–8) | (0–3) | (4–8) | (0–4) | (0.72–1) | (0.69–1) | (0.50–1) | (0.62–1) | (0.67–1) |
| | L1O | CC, n = 32 SU, n = 32 | 29 | 3 | 27 | 5 | 0.86 | 0.88 | 0.84 | 0.91 | 0.9 |
| | L1O | CAARMS = 0, n = 16 CAARMS>0, n = 16 | 10 | 6 | 3 | 13 | 0.4 | 0.41 | 0.19 | 0.62 | 0.33 |
| SVM | HO | CC, n = 8 | 8 | 0 | 5 | 3 | 0.92 | 0.81 | 0.62 | 1 | 1 |
| | | SU, n = 8 | (6–8) | (0–2) | (3–8) | (0–5) | (0.77–1) | (0.62–0.97) | (0.38–1) | (0.75–1) | (0.71–1) |
| | L1O | CC, n = 32 SU, n = 32 | 31 | 1 | 21 | 11 | 0.91 | 0.81 | 0.66 | 0.97 | 0.95 |
| | L1O | CAARMS = 0, n = 16 CAARMS>0, n = 16 | 9 | 7 | 3 | 13 | 0.4 | 0.38 | 0.19 | 0.56 | 0.3 |
| BT | HO | CC, n = 8 | 7 | 1 | 7 | 1 | 0.89 | 0.81 | 0.88 | 0.88 | 0.83 |
| | | SU, n = 8 | (4–8) | (0–4) | (4–8) | (0–4) | (0.73–1) | (0.62–0.94) | (0.5–1) | (0.5–1) | (0.6–1) |
| | L1O | CC, n = 32 SU, n = 32 | 26 | 6 | 26 | 6 | 0.88 | 0.81 | 0.81 | 0.81 | 0.81 |
| | L1O | CAARMS = 0, n = 16 CAARMS>0, n = 16 | 15 | 1 | 2 | 14 | 0.49 | 0.53 | 0.12 | 0.94 | 0.67 |
| LD | HO | CC, n = 8 | 7 | 1 | 5 | 3 | 0.84 | 0.75 | 0.62 | 0.88 | 0.8 |
| | | SU, n = 8 | (4–8) | (0–4) | (3–8) | (0–5) | (0.61–1) | (0.56–0.94) | (0.38–1) | (0.5–1) | (0.56–1) |
| | L1O | CC, n = 32 SU, n = 32 | 27 | 5 | 21 | 11 | 0.84 | 0.75 | 0.66 | 0.84 | 0.81 |
| | L1O | CAARMS = 0, n = 16 CAARMS>0, n = 16 | 6 | 10 | 4 | 12 | 0.28 | 0.31 | 0.25 | 0.38 | 0.29 |
| LD3 | HO | CC, n = 8 | 8 | 0 | 6 | 2 | 0.92 | 0.88 | 0.75 | 1 | 1 |
| | | SU, n = 8 | (6–8) | (0–2) | (4–8) | (0–4) | (0.75–1) | (0.69–1) | (0.5–1) | (0.75–1) | (0.71–1) |
| | L1O | CC, n = 32 SU, n = 32 | 30 | 2 | 27 | 5 | 0.91 | 0.89 | 0.84 | 0.94 | 0.93 |
| | L1O | CAARMS = 0, n = 16 CAARMS>0, n = 16 | 13 | 3 | 3 | 13 | 0.53 | 0.5 | 0.19 | 0.81 | 0.5 |
| LD2 | HO | CC, n = 8 | 7 | 1 | 5 | 3 | 0.91 | 0.81 | 0.62 | 0.88 | 0.88 |
| | | SU, n = 8 | (5–8) | (0–3) | (3–8) | (0–5) | (0.75–1) | (0.62–0.94) | (0.38–1) | (0.62–1) | (0.62–1) |
| | L1O | CC, n = 32 SU, n = 32 | 30 | 2 | 22 | 10 | 0.88 | 0.81 | 0.69 | 0.94 | 0.92 |
| | L1O | CAARMS = 0, n = 16 CAARMS>0, n = 16 | 13 | 3 | 4 | 12 | 0.46 | 0.53 | 0.25 | 0.81 | 0.57 |

Note: TN- true negative, FP–false positive, TP–true positive, FN–false negative. AUC–area under receiver operating characteristic (ROC) curve. For the HO we show median and (2.5–97.5 centiles) based on the 1000 folds.

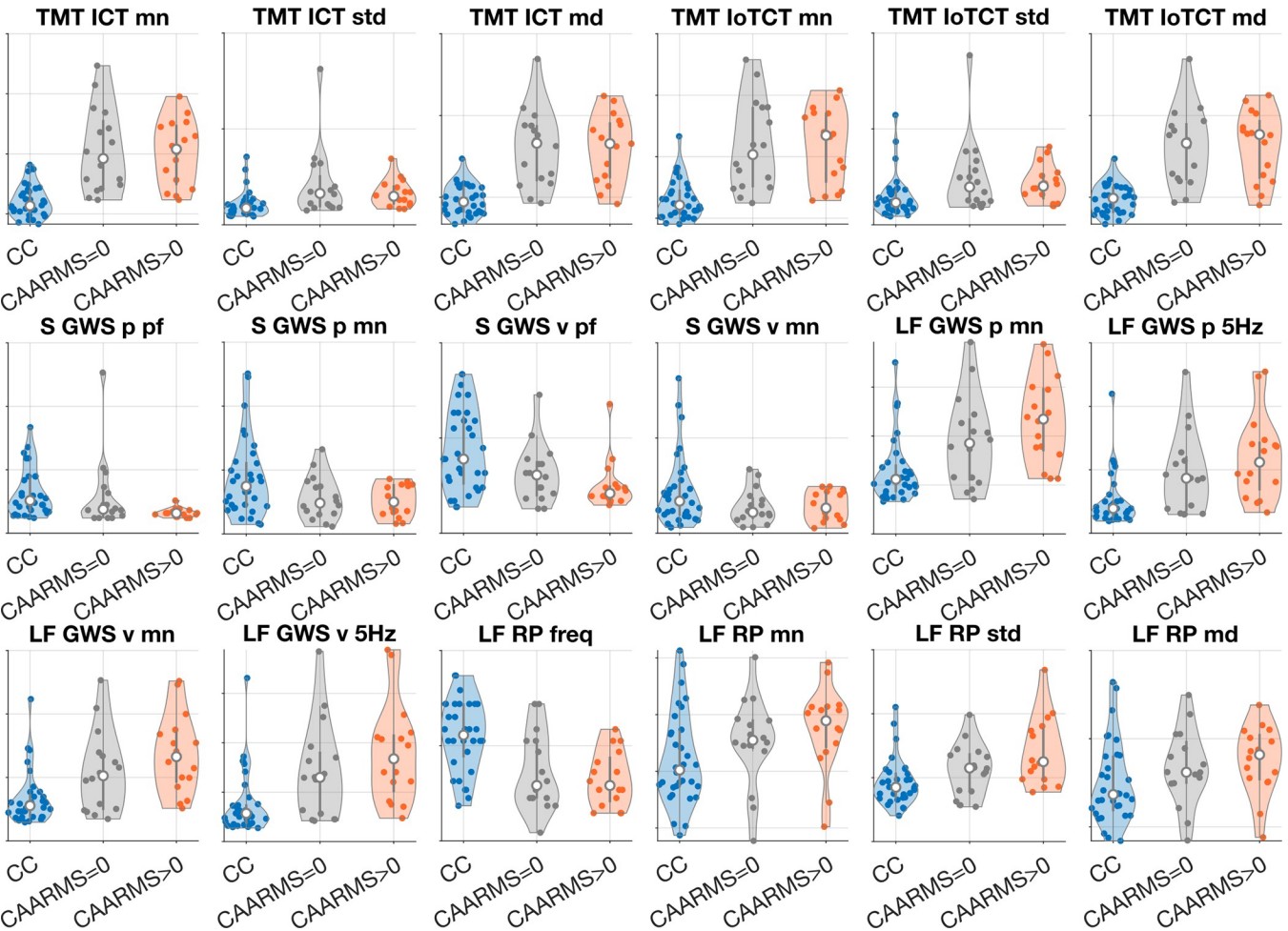

**Fig 3. Distributions of summary statistics measure values in CC (blue) and two groups of SU, CAARMS = 0 (gray) and CAARMS>0 (orange).** Violin plots illustrate distributions of the values, white dot shows median, gray vertical bar shows IQR (middle 50% of values), scatterplots in each violin plot show all individual values. In all plots y-axis shows z-scored values in arbitrary units.

the SU CAARMS = 0 vs SU CAARMS > 0 groups. Interestingly, we observed that the values of most features in the SU CAARMS > 0 cohort differ more from the CC compared to the difference between SU CAARMS = 0 cohort and CC; see Fig 3 and Table 5. Moreover, we observed that the effect size (Cliff's delta) is overall higher for the features from the MG task while remaining relatively unchanged for TMT. The statistical significance of the difference observed in the pattern of change between the SU CAARMS = 0 vs SU CAARMS > 0 groups was investigated using a bootstrap test with 10000 random splits of the SU cohort of observing simultaneous low change in effect sizes for TMT (smaller than the median change of the 6 TMT summary statistics measures), mixed change in effect sizes for Solo MG task (larger than the median change of the 4 Solo MG summary statistics measures) and large change in effect sizes for Leader-Follower MG task (larger than the median change of the 8 Leader-Follower MG summary statistics measures). We found this to be statistically significant with $p < 0.0031$.

## Discussion

We presented results of a feasibility study in which we investigated the potential for employing digital behavioural tests in healthcare practice for stratification of specialist teams for early psychosis

**Table 5. Table of difference between summary statistics (point) measures in CC and SU (combined), CC and SU with CAARMS = 0 and CC and SU with CAARMS>0.**

| Feature | CC vs. SU | | CC vs. SU CAARMS = 0 | | CC vs. CAARMS > 0 | | $\|es_{>0} - es_{=0}\|$ |
|---|---|---|---|---|---|---|---|
| | Effect size | p-value | Effect size | p-value | Effect size | p-value | |
| TMT ICT mean | 0.81 | <0.0001 | 0.8 | 0.00022 | 0.82 | <0.0001 | 0.02 |
| TMT ICT std | 0.55 | 0.0028 | 0.53 | 0.023 | 0.57 | 0.0058 | 0.039 |
| TMT ICT median | 0.84 | <0.0001 | 0.84 | <0.0001 | 0.85 | <0.0001 | 0.012 |
| TMT IoTCT mean | 0.79 | <0.0001 | 0.77 | 0.00022 | 0.8 | <0.0001 | 0.029 |
| TMT IoTCT std | 0.46 | 0.016 | 0.43 | 0.075 | 0.49 | 0.023 | 0.066 |
| TMT IoTCT median | 0.83 | <0.0001 | 0.84 | <0.0001 | 0.83 | <0.0001 | 0.012 |
| S GWS p pf | 0.42 | 0.0028 | 0.24 | 0.46 | 0.61 | 0.00022 | 0.37 |
| S GWS p mean | 0.34 | 0.016 | 0.32 | 0.2 | 0.37 | 0.023 | 0.047 |
| S GWS v pf | 0.44 | 0.016 | 0.34 | 0.2 | 0.54 | 0.0012 | 0.2 |
| S GWS v mean | 0.23 | 0.58 | 0.25 | 0.31 | 0.21 | 0.31 | 0.043 |
| LF GWS pos. mean | 0.55 | 0.00012 | 0.41 | 0.012 | 0.69 | 0.00022 | 0.27 |
| LF GWS pos. 5HZ | 0.58 | 0.00012 | 0.44 | 0.012 | 0.73 | 0.00022 | 0.29 |
| LF GWS vel. mean | 0.54 | 0.00012 | 0.39 | 0.012 | 0.7 | 0.00022 | 0.32 |
| LF GWS vel. 5HZ | 0.58 | 0.00012 | 0.43 | 0.012 | 0.72 | 0.00022 | 0.29 |
| LF RP frequency | 0.59 | 0.001 | 0.52 | 0.042 | 0.66 | 0.0027 | 0.14 |
| LF RP time mean | 0.32 | 0.0028 | 0.25 | 0.023 | 0.4 | 0.012 | 0.15 |
| LF RP time std | 0.48 | 0.001 | 0.35 | 0.042 | 0.61 | 0.0027 | 0.26 |
| LF RP time median | 0.33 | 0.0028 | 0.27 | 0.042 | 0.39 | 0.012 | 0.12 |

Note: Effect size is quantified with Cliff's delta, presented p-value is of the Kolmogorov-Smirnov test. See also Table 3 and Fig 3. $\|es_{>0} - es_{=0}\|$ is the difference between effect sizes of CC vs. SU CAARMS > 0 and CC vs. SU CAARMS = 0.

(STEP) users and healthy cohorts. Our analysis demonstrated that the two investigated behavioural tests (MG and TMT) can be used to differentiate between STEPs users and healthy cohorts with excellent accuracy AUC>0.84 using any of the five analysed classifiers and two different training-testing splits, 25% hold-out and leave-one-out. Excellent performance of the classifiers is driven by statistically significant and large differences (large effect sizes) in features between the cohorts. Finally, we showed that cheap off-the-shelf equipment (laptop computer, 722.76GBP, and a leap motion sensor, 84.24GBP, prices at mid 2018) can be used to record clinically relevant behavioural data and that digital behavioural tests hold the prospect to aid clinical practice.

We also identified areas that require further research and development. We observed that the behavioural data from the MG and TMT collected in the current study cannot be used to differentiate between service users (SU) without (CAARMS = 0) and SU with at-risk-mental-state (CAARMS = 2) or first episode psychosis (CAARMS = 4). This result might partially reflect limited specificity of the CAARMS assessment, meaning that only 15–22% of individuals with at-risk-mental-state develop a full psychotic disorder within 12 months [54–56]. Another possible limitation is the small number of participants available in our SU cohort. Nonetheless, the fact the SU can be so accurately differentiated (large effect sizes for difference between features in the two cohorts) from the CC confirms that the so-called 'non-cases' among STEPs referral have a range of characteristic behavioural markers and constitute an important clinical cohort that differs from control cohort [57–60]. Moreover, the Specificity = 1 achieved by 3 out of 8 tested methods (kNN, SVM, LD3) means that it most accurately identifies control participants. This is important as misclassification in terms of mental health state in young individuals could have equally serious consequences due to stigma associated with mental health diagnosis [61].

Furthermore, we showed (analysis of the effect sizes) that SU with CAARMS>0 differ more from the CC than SU with CAARMS = 0. This indicates presence of differences between these two cohorts which could be uncovered by means of including additional tasks and additional data modalities. For example, recordings of hand movements during the TMT or recordings of eye-movements during both tasks. Inclusion of eye-movement data could be particularly beneficial since it is demonstrated to have diagnostic potential [24,27]. Additionally, using mechanistic (differential equations) models [62] to combine eye-movements, reaction time and movement data could help to identify people's cognitive strategies e.g., employed to complete neuropsychological tasks [63,64]. Identification of the cognitive strategies and understanding their causal mechanisms would elucidate the role of pathophysiology in perturbed information processing and allow the development of new methodologies for risk and treatment stratification.

Finally, longitudinal studies using digital behavioural tests would be instrumental for understanding how motor coordination and other neurological signs change (decline or improvement) in the course of psychosis and why, as shown by Ferruccio and colleagues [31], they allow to predict its long-term severity.

## Features' importance and classifiers' interpretability

Overall excellent performance of the classifiers can be explained by the large effect sizes of the differences between the cohorts in terms of the features used for classification. The effect sizes (presented in Table 5, column 'Effect size') are directly related to the differences (distance) between feature values and affect performance of all the considered classifiers. This conclusion is directly confirmed by the almost identical performance of the kNN classifier using 3 and 18 features, meaning that the 3 features with the highest effect size capture most of the information contained in the 18 features.

The best performance of the LD3 classifier demonstrates that the SU and CC cohorts are linearly separable. While decrease of performance of the LD2 classifier shows that all 3 tasks are important to discriminate between the cohorts. Table 5, column 'Effect size', shows that features derived from the TMT data are the most important and features from the MG Solo are the least important for classification. Worse performance of LD with 18 features could be a result of theoretical ('curse of dimensionality') [65] or numerical (e.g., separating points in higher dimension might require estimation of more complex decision boundaries) limitations.

## Study limitations

There are two main limitations of the study. First, we did not control the level of education in the two groups and it is know that the performance in TMT is affected by years of education [17]. However, even using only the features from the MG allow to classify the CC and SU with AUC = 90 (0.69–1) and sensitivity = 0.75 (95%CI 0.5–1) and specificity = 0.88 (95%CI 0.62–1); kNN, 25% hold-out and 1000 folds. The second, potential source of bias is the short exposure to antipsychotic medication; less than 4 months. We allowed 4 months of antipsychotic medication in order to facilitate recruitment of participants while minimising potential for manifestation of motor side-effects associated with anti-psychotic drugs [66]. Therefore we anticipate the effect of medication status to be minimal and additionally confound with CAARMS score. In an earlier study we have shown that the obtained classification results are independent from anti-psychotic medication status [18]. Furthermore, a recent study showed that neurological signs (e.g., tests of coordination and balance) and their change over 10 years is likely unrelated to exposure to anti-psychotic drugs [31].

## Feasibility of the real-world implementation

As a part of the study, we collected feedback from the participants and healthcare practitioners regarding acceptability and easiness of use of the tests. 65% of participants found the test acceptable as a part of a routine clinical assessment for risk of psychosis (answer >5 on a 10-point scale, 1 –not at all and 10 –very much, to a question: 'How acceptable would you find the activity as a part of a routine clinical assessment for risk of psychosis?') with only 5 out of 32 participants giving a score <5. 87.5% of participants replied that they would be comfortable taking such a test at home (answer >5 on a 10-point scale to a question: 'Would you be comfortable taking such a test at home?') with 17 out of 32 participants giving a score 9 or 10 and only 2 participants giving a score <5 (all questions and SU answers are available at https://doi.org/10.17605/OSF.IO/RNZYS). The healthcare practitioners (doctors and research nurses) who collected the data from the service users provided oral feedback about the tasks and the data collection set-up, they praised its portability (ability to run the tests at service users' homes) and easy set-up (simply plugging in the USB cables into a laptop computer).

We envisage that most immediate potential future implementations of digital batteries of behavioural tests will be taking place within a healthcare setting, in a clinic or practitioners office). We foresee the tests as part of a decision support tool and to be employed as any other medical test. The data collected during the test will be treated as any other patients' data collected during medical examination in terms of ethics and privacy. We believe that augmenting neuropsychological evaluation by means of digital batteries of behavioural test could be saving clinicians time for meaningful conversation with a help-seeking service user. We hope that the time saved by the tests, together with their demonstrated accuracy will help to humanise and destigmatise mental health diagnosis. We are aware that future deployment of such technology would require careful consideration of privacy and related ethical issues, which is beyond the scope of this paper.

## Implications for clinical and research practice

Our findings reinforce the benefits of digital behavioural test and quantitative analysis of their results and their potential for being used as a mobile assessment platform; assessable in home settings as well [67]. Cheap, portable off-the-shelf equipment allows the assessment to take place in a range of indoor locations, while automatic data collection greatly simplifies the necessity for training of clinical personnel.

Digital behavioural tests would benefit research on prognostic instruments for psychosis. Recent review [54] identified heterogeneity in recruitment strategies for high-risk services as one of the factors limiting development of prognostic instruments for psychosis. Digital behavioural test could alleviate this limitation by stratifying and enabling the identification of more homogeneous clinical high-risk populations.

Finally, with further development standardised digital test batteries could supplement and augment neuropsychiatric/ neurological tests making them quicker and easier to apply in routine clinical practice. This would have wide ranging implications for home health, care-coordination and care referral. Therefore, future work should focus on identification of optimal set of tests for establishing standardised digital batteries of behavioural tests and their optimal technological implementations. Such innovative and cost-effective testing methods have the potential to be extended beyond STEPs users' stratification [68] and would facilitate rapid assessments (in clinic or at home) for all people referred to mental health early intervention services [69,70], improving their quality of life and wellbeing.

## Acknowledgments

The research team would like to thank all the STEPs users and control participants who generously shared their time and took part in the project. This study would not be possible without them.

## Author Contributions

**Conceptualization:** Piotr Słowiński, Sarah Sullivan, Tobit Emmens, Philip Self, Jane Wileman, Anke Karl, Krasimira Tsaneva-Atanasova.

**Data curation:** Piotr Słowiński, Alexander White, Sian Lison, Jane Wileman.

**Formal analysis:** Piotr Słowiński.

**Funding acquisition:** Piotr Słowiński, Sarah Sullivan, Tobit Emmens, Philip Self, Krasimira Tsaneva-Atanasova.

**Investigation:** Piotr Słowiński, Alexander White, Krasimira Tsaneva-Atanasova.

**Methodology:** Piotr Słowiński, Sian Lison, Sarah Sullivan, Tobit Emmens, Philip Self, Jane Wileman, Anke Karl, Krasimira Tsaneva-Atanasova.

**Project administration:** Piotr Słowiński, Sian Lison, Sarah Sullivan, Tobit Emmens, Jane Wileman, Krasimira Tsaneva-Atanasova.

**Resources:** Piotr Słowiński, Tobit Emmens, Anke Karl, Krasimira Tsaneva-Atanasova.

**Software:** Piotr Słowiński.

**Supervision:** Piotr Słowiński, Krasimira Tsaneva-Atanasova.

**Validation:** Piotr Słowiński, Krasimira Tsaneva-Atanasova.

**Visualization:** Piotr Słowiński.

**Writing – original draft:** Piotr Słowiński, Alexander White, Krasimira Tsaneva-Atanasova.

**Writing – review & editing:** Piotr Słowiński, Alexander White, Sian Lison, Sarah Sullivan, Tobit Emmens, Philip Self, Jane Wileman, Anke Karl, Krasimira Tsaneva-Atanasova.

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
