## [Decision Letter · Decision Letter 0]

10 May 2023

PDIG-D-23-00071

Digital behavioural tests as diagnostic aid for psychosis

PLOS Digital Health

Dear Dr. Słowiński,

Thank you for submitting your manuscript to PLOS Digital Health. After careful consideration, we feel that it has merit but does not fully meet PLOS Digital Health's publication criteria as it currently stands. Therefore, we invite you to submit a revised version of the manuscript that addresses the points raised during the review process.

Specifically, comment on the (lack of) generalisability of the findings and the model choice. In addition to the reviewers' comments, please revise the title of the manuscript to adequately reflect the study design and small sample size.

Please submit your revised manuscript within 30 days Jun 09 2023 11:59PM. If you will need more time than this to complete your revisions, please reply to this message or contact the journal office at digitalhealth@plos.org. Please include the following items when submitting your revised manuscript:

We look forward to receiving your revised manuscript.

Kind regards,

Laura M. König

Academic Editor

PLOS Digital Health

Journal Requirements:

2. We ask that a manuscript source file is provided at Revision. Please upload your manuscript file as a .doc, .docx, .rtf or .tex.

3. Please provide separate figure files in .tif or .eps format only and remove any figures embedded in your manuscript file. Please also ensure that all files are under our size limit of 10MB.

Additional Editor Comments (if provided):

Reviewers' comments:

Reviewer's Responses to Questions

**Comments to the Author**

1. Does this manuscript meet PLOS Digital Health’s publication criteria? Is the manuscript technically sound, and do the data support the conclusions? The manuscript must describe methodologically and ethically rigorous research with conclusions that are appropriately drawn based on the data presented.

Reviewer #1: Yes

Reviewer #2: Yes

2. Has the statistical analysis been performed appropriately and rigorously?

Reviewer #1: Yes

Reviewer #2: Yes

3. Have the authors made all data underlying the findings in their manuscript fully available (please refer to the Data Availability Statement at the start of the manuscript PDF file)?

Reviewer #1: Yes

Reviewer #2: Yes

4. Is the manuscript presented in an intelligible fashion and written in standard English?

Reviewer #1: Yes

Reviewer #2: Yes

5. Review Comments to the Author

Reviewer #1: The topic is relevant as psychotic illness is rising globally. The authors do a commendable job of trying innovative and cost effective testing methods to timely identify the illness by the general practitioners and consequent referrals to treat it.

Major issues with this study stem from its generalizability, due to very small sample size of cohorts. Also, the mean age is at the lower end, which means the samples may not have much representation of the older population. Also, looks like participants with only English as the first language are selected in the studies. Recommend that authors provide the evidence of data sufficiency and generalizability of the results over the age, education level, and language. With the limited number of participants in the cohort, the spread over regions might be too thin, data is not available to conclude this.

Given above limitations, the pilot using the methods described is interesting, but need broader testing before putting into practice. Given the implications of this study on Home Health and care-coordination and care referral, authors are encouraged to enrich the implications section.

Reviewer #2: The authors proposed to use digital behavioural tests for early psychosis prediction. They extracted 18 features derived from the ‘mirror game’ (MG) and trail making task (TMT) and employed k-nearest neighbours (kNN) classifier for the prediction. After the inclusion and exclusion criteria the number of the positive and negative cases were 32 each. Therefore, they employed 25% hold-out and 1000 folds for the evaluation.

Strength:

1, The prediction problem was justified and well motivated.

2, The methodology was sound and the results looked reasonable.

Weakness:

1, The data size of 64 was a unfortunate limitation of this study. The generalization of the model could be doubtful.

2, Only one model was presented. The authors could try a number more standard machine learning algorithms such as SVM.

3, Model explainability was not explored. What were the key features that affected the model the most in its decision making?

Suggestions:

1, The authors are encouraged to try a number of standard machine learning algorithms such as SVM.

2, Explore model learned weights or other methods to improve model explainability. 

3, Any reporting of experiments or findings should be in past tense.

6. PLOS authors have the option to publish the peer review history of their article (what does this mean?). If published, this will include your full peer review and any attached files.

**Do you want your identity to be public for this peer review?** For information about this choice, including consent withdrawal, please see our Privacy Policy.

Reviewer #1: Yes: Pankaj Jain

Reviewer #2: No

---

## [Decision Letter · Decision Letter 1]

20 Jul 2023

PDIG-D-23-00071R1

The potential of digital behavioural tests as a diagnostic aid for psychosis

PLOS Digital Health

Dear Dr. Słowiński,

Thank you for submitting your manuscript to PLOS Digital Health. After careful consideration, we feel that it has merit but does not fully meet PLOS Digital Health's publication criteria as it currently stands. Therefore, we invite you to submit a revised version of the manuscript that addresses the points raised during the review process.

Please submit your revised manuscript within 60 days Sep 18 2023 11:59PM. If you will need more time than this to complete your revisions, please reply to this message or contact the journal office at digitalhealth@plos.org. Please include the following items when submitting your revised manuscript:

We look forward to receiving your revised manuscript.

Kind regards,

Laura M. König

Academic Editor

PLOS Digital Health

Journal Requirements:

Additional Editor Comments (if provided):

Reviewers' comments:

Reviewer's Responses to Questions

**Comments to the Author**

1. If the authors have adequately addressed your comments raised in a previous round of review and you feel that this manuscript is now acceptable for publication, you may indicate that here to bypass the “Comments to the Author” section, enter your conflict of interest statement in the “Confidential to Editor” section, and submit your "Accept" recommendation.

Reviewer #2: All comments have been addressed

Reviewer #3: All comments have been addressed

2. Does this manuscript meet PLOS Digital Health’s publication criteria? Is the manuscript technically sound, and do the data support the conclusions? The manuscript must describe methodologically and ethically rigorous research with conclusions that are appropriately drawn based on the data presented.

Reviewer #2: Yes

Reviewer #3: Yes

3. Has the statistical analysis been performed appropriately and rigorously?

Reviewer #2: Yes

Reviewer #3: Yes

4. Have the authors made all data underlying the findings in their manuscript fully available (please refer to the Data Availability Statement at the start of the manuscript PDF file)?

Reviewer #2: No

Reviewer #3: Yes

5. Is the manuscript presented in an intelligible fashion and written in standard English?

Reviewer #2: Yes

Reviewer #3: Yes

6. Review Comments to the Author

Reviewer #2: The authors have addressed most issues. Particularly, they added the Cliff's Delta effect size to demonstrate the strength of relationship between the dependent and independent variables, which is a sample size independent measure. I am not sure if it can explain away the issue of generalizability of machine learning models on a small data set.

Reviewer #3: Small Sample Size: The study uses a small sample size (n=32 for both groups). This restricts the generalizability of the results and may not represent the heterogeneity of individuals with psychotic illnesses.

Overfitting Risk: The study doesn't mention cross-validation across multiple datasets, only a hold-out set. This could result in overfitting and may fail to generalize to unseen data.

Limited Test Effectiveness: The mirror game and trail making task were found unsuitable in isolation to differentiate between critical subgroups (those at risk or with first episode psychosis). Given the variability in psychotic states, additional or different tests might be needed.

Feasibility Concerns: Implementation feasibility in a real-world setting isn't discussed. This includes patient comfort with technology, practitioner technological literacy, and the true cost of equipment and implementation.

Lack of Validation Against Clinical Diagnosis: It's not clear if the results were validated against a clinical diagnosis made by a psychiatrist. Without this, the applicability of results in a clinical setting is uncertain.

Privacy and Ethics: Digital tests often require collection of sensitive patient data, which raises privacy and ethical concerns not addressed in the study.

7. PLOS authors have the option to publish the peer review history of their article (what does this mean?). If published, this will include your full peer review and any attached files.

**Do you want your identity to be public for this peer review?** For information about this choice, including consent withdrawal, please see our Privacy Policy. 

Reviewer #2: No

Reviewer #3: No

---

## [Editor Report · Decision Letter 2]

29 Jul 2023

The potential of digital behavioural tests as a diagnostic aid for psychosis

PDIG-D-23-00071R2

Dear Dr Słowiński,

We are pleased to inform you that your manuscript 'The potential of digital behavioural tests as a diagnostic aid for psychosis' has been provisionally accepted for publication in PLOS Digital Health.

Best regards,

Laura M. König

Academic Editor

PLOS Digital Health